# Measuring socioeconomic inequalities in prenatal HIV test service uptake for prevention of mother to child transmission of HIV in East Africa: A decomposition analysis

Feleke Hailemichael Astawesegn[1,2]*, Elizabeth Conroy[1], Haider Mannan[1], Virginia Stulz[3]

1 Translational Health Research Institute (THRI), School of Medicine, Western Sydney University, Penrith, NSW, Australia, 2 School of Public Health, College of Medicine and Health Sciences, Hawassa University, Hawassa, Ethiopia, 3 School of Nursing and Midwifery Centre for Nursing and Midwifery Research, Western Sydney University, Nepean, Hospital 1st level Court Building, Kingswood, NSW, Australia

* felekeh86@gmail.com

## Abstract

### Background

Despite efforts made towards the elimination of mother-to-child HIV transmission, socioeconomic inequality in prenatal HIV test uptake in East Africa is not well understood. Therefore, this study aimed at measuring socioeconomic inequalities in prenatal HIV test uptake and explaining its main determinants in East Africa

### Method

We analysed a total weighted sample of 45,476 women aged 15–49 years who birthed in the two years preceding the survey. The study used the most recent DHS data from ten East African countries (Burundi, Comoros, Ethiopia, Kenya, Malawi, Mozambique, Rwanda, Uganda, Zambia, and Zimbabwe). The socioeconomic inequality in prenatal HIV test uptake was measured by the concentration index and illustrated by the concentration curve. Then, regression based Erreygers decomposition method was applied to quantify the contribution of socioeconomic factors to inequalities of prenatal HIV test uptake in East Africa.

### Results

The concentration index for prenatal HIV test uptake indicates that utilization of this service was concentrated in higher socio-economic groups with it being 15.94% higher among these groups in entire East Africa (p <0.001), 40.33% higher in Ethiopia (p <0.001) which was the highest and only 1.87% higher in Rwanda (p <0.01) which was the lowest. The decomposition analysis revealed that household wealth index (38.99%) followed by maternal education (13.69%), place of residence (11.78%), partner education (8.24%), watching television (7.32%), listening to the radio (7.11%) and reading newsletters (2.90%) made the largest contribution to socioeconomic inequality in prenatal HIV test in East Africa.

**Data Availability Statement:** Data used for this study is owned by measure DHS/ICF International. However, the data are available to download after

online registration for research purposes via https://www.dhsprogram.com/data/available-datasets.cfm. The Information on the HIV testing during antenatal care and before birth for the most recent live birth preceding the survey for each woman in the sample was found in the DHS dataset individual record (IR) file. The variables code and descriptions were (v840, Information about HIV testing during antenatal care), (v841, Information about receiving HIV test results during antenatal care), (v840a, Information about HIV testing before birth) and (v841a, Information about receiving HIV test results before birth). For this study, the authors confirm that they obtained approval from Measure DHS/ ICF International.

**Funding:** The author(s) received no specific funding for this work.

**Competing interests:** The authors have declared that no competing interests exist.

**Abbreviations:** AIDS, acquired immune deficiency syndrome; ART, Antiretroviral therapy; CI, Concentration index; DHS, Demographic and Health Survey; ECI, Erreygers Normalized Concentration Index; HIV, Human immune virus; PMTCT, Prevention of mother-to-child transmission of HIV; SDGs, Sustainable Development Goals; SES, Socio-economic status; UNAIDS, United Nations Programme on HIV/AIDS; WHO, World Health Organization.

## Conclusion

In this study, pro-rich inequality in the utilization of prenatal HIV tests was evident. The decomposition analysis findings suggest that policymakers should focus on improving household wealth, educational attainment, and awareness of mother-to-child transmission of HIV (MTCT) through various media outlets targeting disadvantaged sub-groups.

## Introduction

HIV remains to be a major global public health problem [1]. Across the world, approximately 37.7 million people were living with HIV of these 1.7 million were children aged 0–14 years at the end of 2020 [2, 3]. The burden of the epidemic continues to vary considerably between countries and regions. Sub-Saharan Africa remains most severely affected region accounting for more than two-thirds of the people living with HIV followed by Asia and the Pacific world-wide [3, 4].

The spread of HIV/AIDS disproportionately affected women in sub-Saharan Africa (SSA) [5] due to poverty, political inequalities, socially constructed gender inequalities and inadequate coverage of health care services [5, 6]. Moreover, women are at a higher risk of acquiring HIV during pregnant or breastfeeding period [7–9] and those women who acquire HIV during pregnancy or the postpartum period are more likely to pass the infection on to their offspring [10, 11]. As a result, vertical transmission of HIV remains a public health challenge in the region. Evidence showed that 90% of children age less than 15 years who lived with HIV/AIDS found in SSA [12]. Furthermore, in 2018, 61% of new HIV infections [13] and 73% of the deaths occurred in SSA alone [14]. Specifically, 67% of children living with HIV are found in Eastern and Southern Africa [4]. Hence, preventing mother-to-child-transmission of HIV (PMTCT) is an essential HIV/AIDS infection control strategy amongst children in the region.

Early identification of HIV-positive pregnant women during pregnancy, birth, or the postpartum period is vital to prevent the risk of MTCT of HIV as it's a gateway to prevention of HIV transmission, treatment, care and support services [15]. Provider-initiated HIV testing and counselling (PITC) (sometimes referred to as opt-out HIV testing) is recommended during antenatal care (with repeat testing at birth for those who initially test negative) to detect HIV-positive mothers [16]. Testing remains voluntary and the woman is given the option to 'opt-out' of testing [17]. Those women who test positive for HIV are supposed to be immediately provided with Anti-Retroviral Therapy (ART) both to decrease the risk of mother-to-child HIV transmission (MTCT) and to improve maternal health [17]. Evidence has shown that the overall prenatal HIV test service uptake was 80.8% in World Health Organization regions of East African countries (Burundi, Comoros, Ethiopia, Kenya, Malawi, Mozambique, Rwanda, Uganda, Zambia, and Zimbabwe). The lowest prenatal HIV testing was measured in Comoros (17%) and highest was in Rwanda (97.9%) [18], and most of these countries were below the minimum 95% required to enable 95% of HIV-positive pregnant women to have access to ART [19–21].

Various studies have examined the relationship between HIV testing and socioeconomic factors and have indicated that the utilization of HIV testing is highly associated with socioeconomic status (SES) such as differences in age, marital status, educational status, and place of residence [22–30]. These differences associated with socioeconomic and demographic factors are an indication of the presence of socioeconomic inequalities in prenatal HIV test service uptake [31, 32]. However, the level of socioeconomic inequalities and the contribution of each

socio-economic and demographic factor on prenatal HIV tests for prevention of mother-to-child transmission of HIV (PMTCT) in East Africa has not been empirically investigated.

Measuring socioeconomic inequalities in health service use can guide the focus of policy-makers and assist stakeholders to prioritize specific health care needs [33]. Improving socio-economic inequalities in healthcare utilization has become one of the most important priorities or a shared goal amongst many policymakers, researchers, and public health practitioners who strive to improve health systems [34]. For instance, the principle of universal health coverage (UHC) states that individuals with equal needs should utilize equal healthcare services without any financial hardship [34]. The United Nation sustainable development goals (SDGs), specifically SDG 3, calls for ensuring healthy lives and promoting well-being for all, whilst SDG 10 calls for reducing inequality within and between countries to promote the inclusion and empowerment of all [35]. Under these broad goals, UNAIDS has initiated a new approach "End Inequalities, to End AIDS Epidemic" [36]. It is a global AIDS strategy to close the gaps that are preventing progress towards ending HIV/AIDS by reducing inequalities in access to treatment and care. Hence, UNAIDS recommend that every country to sets out evidence-based priority actions and targets to reduce inequalities and end AIDS as a public health threat by 2030 [14, 36, 37].

Therefore, the purpose of this study was to estimate the level of socioeconomic inequality in the uptake of prenatal HIV test services for PMTCT in East African countries using concentration index decomposition approach. Firstly, with concentration index (CI) we measure the extent/magnitude of socioeconomic inequalities in prenatal HIV test service utilization [38, 39]. Secondly, with the regression-based decomposition approach, we quantified the contributions of the socio-economic factors for the observed inequalities in the uptake of prenatal HIV test services. Hence, quantifying prenatal HIV test service uptake inequalities that exist, identification and quantification of covariates that explain the inequalities will assist policy maker to prioritize and develop tailored policies, strategies, and actions aimed at reducing the existing inequalities between the poor and rich. It can be also helpful to assess/evaluate the extent to which PMTCT services are better targeted toward underprivileged in Eastern Africa countries.

## Methods and materials

### Study design and data sources

Data were extracted from the most recent Demographic and Health Surveys (DHS), consisting ten East Africa countries with complete data for the period from 2011 to 2018 [40, 41]. Countries were Burundi (DHS,2016–17), Comoros (DHS,2012), Ethiopia (DHS,2016), Kenya (DHS,2014), Malawi (DHS,2015–16), Mozambique (DHS,2011), Rwanda (DHS,2014–15), Uganda (DHS,2016), Zambia (DHS,2018), and Zimbabwe (DHS,2015).

The DHS are based on nationally representative samples that provide data for a wide range of monitoring and impact evaluation indicators in the areas of population, health, and nutrition [42] and therefore, its an important source of data on health of families in developing countries.

The DHS employed a cross-sectional study design with a stratified two-stage sampling strategy. First, each country was divided into enumeration areas (clusters) based on the census frames in the country, and then, households were randomly selected within each cluster. Furthermore, since the DHS surveys were intended to address household-based health issues, strata for urban and rural households were used for the selection of respondents. The DHS follows a standard procedure of data collection and presentation (similar questionnaires) and uses the same definition of terms in each country. The DHS data were collected by the country-specific department of health and population, in collaboration with Inner City Fund (ICF)

International using standardized household questionnaires. The detailed methodology of the survey design, sample selection, survey tools, and data collection are described elsewhere [43, 44].

Pooled DHS datasets from ten East African countries were structured by creating a country-specific cluster and country-specific strata. Hence, in this study, we used a total weighted sample of 45,476 women aged 15–49 years who birthed in the two years preceding the survey. The information on the HIV testing during antenatal care and birth in the two years preceding the survey for each woman in the sample was found in the women's individual records of DHS to measure socioeconomic inequalities in prenatal HIV test service uptake for PMTCT

This study used data from the Measure DHS/ ICF International which is secondary in nature. The data were anonymous and are available to apply for online. Therefore, Approval was received from DHS/ ICF International to use.

## Variables

**Outcome variable.**   The outcome variable in this study included socioeconomic inequality in prenatal HIV test service uptake. Prenatal HIV testing was defined as the proportion of women who tested for HIV and received the test result during antenatal care or before birth [45]. The variable was coded as "1" if a woman was tested for HIV and also received the HIV test result otherwise coded as "0" if the woman did not test for HIV or tested but did not receive the test result during antenatal care or before birth.

**Socioeconomic and demographic factors.**   Based on previously published literature from low and middle-income countries [18, 19, 24], the following socioeconomic and demographic factors were selected for this study and examined: country of participants, place of residence (urban and rural), maternal age (categorized as 15–24 years, 25–34 years, and 35–49 years), and maternal and partner educational level (categorized as no education, primary education, secondary and higher education). The sex of the household head was coded as 'male' if the participants lived in the male-dominated household, or 'female' if otherwise. The household size was classified as 1–3, 4–6, or ≥7 members and employment status were categorized as not working, formal employment, and non-formal employment. Women's access to mass media was categorized as yes or no in relation to access to the radio, watching television, and reading magazines/newspapers.

**Socioeconomic status measure.**   The calculation of concentration indices (CI) requires a single indicator to capture respondents' SES. Hence, the DHS household wealth index was used as a measure of the SES of mothers in East Africa [46]. The household wealth index stands out to be the most appropriate measure of SES for national surveys in comparison with direct measures of living standards such as income, consumption, or expenditure [47]. The wealth index was constructed via principal components analysis (PCA) [48] using data such as household assets (e.g. radio, televisions, refrigerators, farmland, farm animals) [48, 49], housing characteristics (e.g. type of water access, type of flooring), and access to basic services (e.g. electricity supply, source of drinking water and sanitation facilities) [46]. PCA is a multivariate statistical method that is widely used as a data reduction technique [48]. The detailed methodology of the wealth index construction is described elsewhere [46, 49]. In this study, wealth index score was divided into quintiles, each category comprising 20% of the population. The lowest 20% quintile was assigned to the poorest households, the next 20% quintile to generally poor households, followed by another 20% quintile for the middle-class households, and finally the top 40% quintile for the wealthier and wealthiest households.

**Analytical strategy.**   The analytical approach for this study was undertaken in three stages. Firstly, sample characteristics and prevalence of prenatal HIV test service uptake were

described using frequencies and percentages. Secondly, concentration indexes (CIs) were determined to examine the extent of socioeconomic inequalities in prenatal HIV test service uptake in East Africa. Thirdly, we decomposed the concentration indices to understand the contribution of various factors to inequality. The methods are discussed in detail as follows.

**Concentration index and curve.** Concentration index (CI) has become a popular tool to measure health and health care inequality in the field of health policy and health economics research [50]. It assumes values between -1 and +1 [38]. When the concentration index is positive, it reflects higher uptake of HIV test services tend to the rich. If it is negative, it reflects higher uptake of HIV test services tend to the poor. In the absence of any inequalities, the value of the concentration index is zero [38]. It can be estimated as:

$$CI = \frac{2}{\mu} \text{cov}(yi, Ri) \qquad [1]$$

Where CI is the concentration index for health service use (prenatal HIV test service uptake in this case), yi is health service use for individual i, μ is the mean of health service use, and $R_i$ is the fractional rank or asset score of the $i^{th}$ individual in the living standards distribution/ socio-economic rank with $i = 1$ for the poorest and $i = $ n for the wealthiest. The concentration index depends only on the relationship between the health variable ($yi$) and the rank of the living standard variable (Ri) and not on the variation of the living standard variable itself.

When the health variable of interest is dichotomous, the concentration index is not bounded within the range of (–1,1) [51]. The lower bound is then equal to $\mu - 1 + \left(\frac{1}{n}\right)$, the upper bound is equal to $1 - \mu + \left(\frac{1}{n}\right)$ [38]. For large samples, the $\left(\frac{1}{n}\right)$ terms vanish, and the maximum and the minimum values tend to μ−1 and 1−μ respectively [51]. To consider the bounded nature of prenatal HIV test service uptake, Erreygers (2009) proposed a modified version of the concentration index called Erreygers normalized concentration index (ECI) [52]. This is defined as:

$$ECI = 4 * \mu * CI \qquad [2]$$

where ECI is Erreygers concentration index, CI is the generalized concentration index and μ is the mean of prenatal HIV test service uptake. The Erreygers normalized concentration index (ECI) method was also applied to decompose prenatal HIV test service utilization inequality in this study [39, 52]. Furthermore, with concentration curve, the cumulative proportion of women ranked by SES (on the x-axis) against the cumulative proportion of prenatal HIV test service uptake (on the y-axis) were plotted. If the concentration curve lays below the 45-degree line, it suggests prenatal HIV test service use is concentrated more amongst women from rich households and vice versa.

## Decomposition of the concentration index

Decomposition of the healthcare inequality relies on the assumption that healthcare is a linear function of the outcome variables [38]. However, since the outcome variable of this study is binary, an appropriate statistical technique for non-linear settings is needed. According to O'Donnell, et al. [38], a linear approximation when dealing with a discrete change from 0 to 1 is to use marginal or partial effects (dh/dx) estimates as follows:

$$y = \sum_k \beta_k{}^m x_k + e \qquad [3]$$

Where $\beta_k{}^m$ is the marginal effects (dy/dx) of each xk; $e$ indicates the error term generated by the linear approximation. In this study, we performed the decomposition analysis using a

generalized linear model (GLM) with a binomial distribution and a logit link function as a linear approximation to capture the partial effects of socioeconomic factors on prenatal HIV test service uptake [53]. GLM is a suitable regression to provide consistent results for the decomposition of binary outcomes regardless of the choice of the reference category [53]. Therefore, given the relationship between $y$ and $x_k$ in Eq [3], we decomposed the concentration index of prenatal HIV test service uptake y(CI) into its contributory factors as follows

$$CI = \sum_K \left(\frac{\beta_k{}^m \bar{x}_k}{\mu}\right) CI_k + \frac{GCe}{\mu} \qquad [4]$$

In this expression, $\bar{x}_k$ is the means of explanatory variables, $\beta_k{}^m$ is the patrial effect on explanatory variable xk (d$y$/d$xk$), $CI_k$ is the concentration index for determinant $x_k$, and $GCIe$ is the generalized concentration index for the error term.

In summary, the contribution of the determinants is calculated in four steps. Firstly, the regression model of the health outcome variable is performed for all $x_k$ to obtain the marginal effects of determinants ($\beta_k{}^m$), which demonstrate the associations between the determinants and prenatal HIV test service uptake (positive signs indicate positive associations whilst negative signs indicate negative associations). Secondly, the elasticity of the health variables was calculated for each x ($x_k$), which is the sensitivity of prenatal HIV test service uptake to changes in the determinants $\left(\frac{\beta_k{}^m \bar{x}_k}{\mu}\right)$. It denotes the change in the dependent variable (socioeconomic inequality in prenatal HIV test service uptake in this case) associated with a one-unit change in the explanatory variables. Thirdly, the CIs are calculated for the prenatal HIV test uptake and each explanatory variable ($CI_k$). Fourthly, the contribution of each explanatory variable to the overall CI is calculated by multiplying the elasticity of each determinant by its concentration index $\left(\left(\frac{\beta_k{}^m \bar{x}_k}{\mu}\right) CI_k\right)$.

In this study, we accounted for the multistage survey design during descriptive, regression, and decomposition analyses by creating country-specific clustering, country-specific strata, and population-level weight. All statistical analyses were conducted using STATA version 14.2 (Stata Corp, College Station, TX, USA).

## Results

### Characteristics of the study participants

In this study, a weighted number of 45,476 women were included in the analysis from ten East African countries (Burundi, Comoros, Ethiopia, Kenya, Malawi, Mozambique, Rwanda, Uganda, Zambia, and Zimbabwe) between the ages of 15–49 years who birthed two years before the survey. Most study participants were from rural areas (n = 35365, 77.8%), aged between 25–34 years (n = 20365, 44.8%), and had educational qualifications of at least primary education (n = 23739, 52.2%). Three quarters of the respondents (n = 34746, 76.4%) were from a male-headed household, (n = 7727, 17.0%) read newsletters, (n = 26282, 57.8%) listened to the radio and (n = 12510, 27.5%) watched television [**Table 1**].

The overall weighted prevalence of prenatal HIV test uptake was 80.8% (95% CI: 79.7–81.7%) with the highest prevalence in Rwanda (97.9%, 95% CI: 97.2–98.3%) and lowest in Comoros (17.0%, 95% CI: 13.9–20.7%) [**Tables 1 and 2**]. Furthermore, the distribution of prenatal HIV test service uptake across countries and study factors is shown in Tables 1 and 2. For instance, women who resided in rural areas of Burundi (n = 4384, 81.0%), Malawi (n = 5131, 76.7%), Rwanda (n = 2612, 80.7%), and Uganda (n = 4185, 70.9%) had the highest proportion of prenatal HIV test services use. Amongst women who attained secondary and

**Table 1. Study participant characteristics and percentage of prenatal HIV test service users for PMTCT by study factors in East African countries (N = 45,476; Demographic and Health Survey, 2011–2018).**

| Characteristics | Total sample (N = 45476) | | Burundi (N = 5412) | | Comoros (N = 1298) | | Ethiopia (N = 4308) | | Kenya (N = 7357) | |
|---|---|---|---|---|---|---|---|---|---|---|
| | N (%) | Yes# n (%) | N (%) | Yes# n (%) | N (%) | Yes#n (%) | N (%) | Yes# n (%) | N (%) | Yes# n (%) |
| Residency | | | | | | | | | | |
| Urban | 10111(22.2) | 9133(20.1) | 489(9) | 471(8.7) | 368(28.4) | 96 (7.4) | 520(12.1) | 410(9.5) | 2618(35.6) | 2530 (34.4) |
| Rural | 35365(77.8) | 27588(60.7) | 4923(91) | 4384(81.0) | 929(71.6) | 125(9.6) | 3788(87.9) | 1069(24.8) | 4739(64.4) | 4306 (58.5) |
| Maternal age | | | | | | | | | | |
| 15–24 | 17049(37.5) | 13973(30.8) | 1407(26) | 1255(23.2) | 404(31.2) | 68 (5.3) | 1260(29.3) | 453 (10.5) | 2864(38.9) | 2688 (36.5) |
| 25–34 | 20365(44.8) | 16536(36.4) | 2742(50.7) | 2481(45.9) | 622(47.9) | 111 (8.5) | 2189(50.9) | 766 (17.8) | 3432(46.7) | 3186 (43.3) |
| 35–49 | 8024(17.7) | 6194(13.6) | 1259(23.3) | 1117(20.6) | 271(20.9) | 42 (3.2) | 854(19.9) | 260 (6.0) | 1061(14.4) | 963 (13.1) |
| Maternal education | | | | | | | | | | |
| No education | 10312(22.7) | 6116(13.5) | 2365(43.7) | 2096(38.7) | 562(43.3) | 64 (4.9) | 2606(60.5) | 612 (14.2) | 834(11.3) | 615 (8.4) |
| Primary | 23739(52.2) | 20035(44.1) | 2398(45.3) | 2155(39.8) | 323(24.9) | 46 (3.5) | 1319(30.6) | 571 (13.3) | 4023(54.7) | 3775 (51.3) |
| Secondary and higher | 11420(25.1) | 10570(23.2) | 650(12) | 603(11.2) | 413(31.8) | 111 (8.6) | 383(8.9) | 296 (6.8) | 2499(34) | 2445(33.2) |
| Maternal occupation | | | | | | | | | | |
| Not working | 15331(37.0) | 11127(26.8) | 388(7.2) | 360 (6.7) | 816(65.1) | 139 (11.1) | 2510(58.8) | 776 (18.2) | 1885(53.4) | 1730 (49.0) |
| Professional work | 6219(15.0) | 5345(12.9) | 489(9.1) | 453 (8.4) | 203(16.2) | 43 (3.4) | 703(16.5) | 331 (7.8) | 577(16.3) | 559 (15.8) |
| Nonprofessional work | 19922(48.0) | 16570(39.9) | 4502(83.7) | 4009(74.5) | 234(18.7) | 21 (1.7) | 1054(24.7) | 349 (8.2) | 1068(30.3) | 988 (28.0) |
| Partner education | | | | | | | | | | |
| No education | 7051(19.6) | 4187(11.6) | 1775(36.2) | 1589(32.3) | 459(36.9) | 39 (3.1) | 1838(45.1) | 461 (11.3) | 304(9.5) | 220 (6.3) |
| Primary | 17778(49.3) | 14214(39.5) | 2532(51.5) | 2270(46.2) | 322(25.9) | 43 (3.5) | 1651(40.5) | 545 (13.4) | 1609(50.6) | 1500 (47.1) |
| Secondary and Higher | 11191(31.1) | 9988(27.7) | 605(12.3) | 559 (11.4) | 462(37.1) | 117(9.4) | 588(14.4) | 393 (9.6) | 1270(39.9) | 1237 (38.9) |
| Household head | | | | | | | | | | |
| Male | 34746(76.4) | 27919(61.4) | 4436(82.0) | 3973(73.4) | 886(68.2) | 159(12.2) | 3725(86.5) | 1275(29.6) | 5435(73.9) | 5079(69.0) |
| Female | 10730(23.6) | 8802(19.4) | 976(18.0) | 882(16.3) | 412(31.8) | 62(4.8) | 583(13.5) | 204(4.7) | 1922(26.1) | 1757(23.9) |
| Household size | | | | | | | | | | |
| 1–3 | 7588(16.7) | 6293(13.8) | 809(14.9) | 724(13.4) | 165(12.7) | 28(2.2) | 537(12.4) | 241(5.6) | 1356(18.4) | 1282(17.4) |
| 4–6 | 22908(50.4) | 18706(41.1) | 2783(51.4) | 2507(46.3) | 581(44.7) | 85(6.6) | 2197(51.0) | 834(19.4) | 3757(51.0) | 3531(48.0) |
| > = 7 | 14980(32.9) | 11722(25.8) | 1820(33.6) | 1624(30.0) | 552(42.5) | 107(8.3) | 1574(36.5) | 404(9.4) | 2244(30.5) | 2022(27.5) |
| Household wealth index | | | | | | | | | | |
| Poorest | 8148(17.9) | 6006(13.2) | 1161(21.5) | 1005(18.6) | 226(17.4) | 11(0.9) | 355(8.2) | 43(1.0) | 812(11.0) | 605(8.2) |
| Poorer | 9159(20.1) | 6965(15.3) | 1157(21.4) | 1025(18.9) | 262(20.2) | 32(2.5) | 887(20.6) | 171(3.9) | 1349(18.3) | 1234(16.8) |
| Middle | 9611(21.1) | 7496(16.5) | 1149(21.2) | 1038(19.2) | 263(20.2) | 42(3.3) | 1225(28.4) | 325(7.5) | 1480(20.1) | 1385(18.8) |
| Wealthier | 9525(21.0) | 7876(17.3) | 1106(20.4) | 999(18.4) | 269(20.7) | 59(4.6) | 1281(29.7) | 504(11.7) | 1614(21.9) | 1550(21.1) |
| Wealthiest | 9032(19.9) | 8378(18.4) | 840(15.5) | 788(14.6) | 279(21.5) | 76(5.8) | 559(13.0) | 436(10.1) | 2102(28.6) | 2062(28.0) |
| Reads newspaper | | | | | | | | | | |
| No | 37733(83.0) | 29553(65.0) | 5181(95.7) | 4643(85.8) | 1105(85.1) | 154 (11.9) | 4004(93) | 1266(29.4) | 5214(70.9) | 4753 (64.6) |
| Yes | 7727(17.0) | 7159(15.8) | 231(4.3) | 212 (3.9) | 193(14.9) | 67 (5.2) | 304(7) | 213 (4.9) | 2141(29.1) | 2081 (28.3) |
| Listens to the radio | | | | | | | | | | |
| No | 19179(42.2) | 14101(31.0) | 3045(56.3) | 2664(49.2) | 649(50) | 92 (7.1) | 3115(72) | 888 (20.6) | 1564(21.3) | 1317(17.9) |
| Yes | 26282(57.8) | 22610(49.7) | 2367(43.7) | 2191(40.5) | 649(50) | 129 (9.9) | 1193(28) | 591 (13.7) | 5790(78.7) | 5516 (75.0) |
| Watches television | | | | | | | | | | |
| No | 32950(72.5) | 25862(56.9) | 5022(92.8) | 4488(82.3) | 444(34.2) | 51 (3.9) | 3494(81.1) | 986 (22.9) | 4243(57.7) | 3813 (51.9) |
| Yes | 12510(27.5) | 10848(23.8) | 390(7.2) | 366 (6.8) | 854(65.8) | 170 (13.1) | 814(18.9) | 493 (11.4) | 3108(43.3) | 3018 (41.1) |
| HIV test uptake (%), 95% CI | 80.8% [CI: 79.7–81.7%] | | 89.7% [CI: 88.4–90.8] | | 17.0% [CI: 13.9–20.7%] | | 34.3% [CI: 30.3–38.6] | | 92.9% [CI: 92.1–93.7] | |

N = Total weighted number of eligible women; Yes#n = Weighted number of women who tested for HIV; % = weighted proportions of women who were tested for HIV by study factors in each East African country.

**Table 2. Study participants characteristics and percentage of prenatal HIV test service users for PMTCT by study factors in East African countries (N = 45,476; Demographic and Health Survey, 2011–2018).**

| Characteristics | Malawi (N = 6693) N (%) | Yes# n (%) | Mozambique (N = 4913) N (%) | Yes# n (%) | Rwanda (N = 3236) N (%) | Yes# n (%) | Uganda (N = 5901) N (%) | Yes# n (%) | Zambia (N = 3905) N (%) | Yes# n (%) | Zimbabwe (N = 2454) N (%) | Yes# n (%) |
|---|---|---|---|---|---|---|---|---|---|---|---|---|
| Residency | | | | | | | | | | | | |
| Urban | 911(13.6) | 848 (12.6) | 1356 (27.6) | 1050 (21.4) | 561(17.3) | 554 (17.1) | 1258 (21.3) | 1214 (20.6) | 1340 (34.3) | 1307 (33.5) | 689(28.1) | 653 (26.6) |
| Rural | 5781 (86.4) | 5131 (76.7) | 3557 (72.4) | 1884 (38.3) | 2675 (82.7) | 2612 (80.7) | 4643 (78.7) | 4185 (70.9) | 2564 (65.7) | 2328 (59.6) | 1765 (71.9) | 1563 (63.7) |
| Maternal age | | | | | | | | | | | | |
| 15–24 | 3083 (46.1) | 2717 (40.6) | 2015 (41.2) | 1223 (24.9) | 871(26.9) | 852(26.3) | 2511 (42.6) | 2290 (38.8) | 1666 (42.7) | 1544 (39.6) | 966(39.4) | 882 (36.0) |
| 25–34 | 2584 (38.7) | 2361 (35.3) | 2000 (40.8) | 1250 (25.5) | 1666 (51.5) | 1634 (50.5) | 2473 (41.9) | 2306 (39.1) | 1514 (38.8) | 1413 (36.2) | 1144 (46.6) | 1029 (41.9) |
| 35–49 | 1018 (15.2) | 897 (13.4) | 882(18) | 455 (9.3) | 699(21.6) | 680(21.0) | 912(15.5) | 799 (13.6) | 724(18.5) | 678(17.4) | 343(14) | 305 (12.4) |
| Maternal education | | | | | | | | | | | | |
| No education | 794(11.9) | 673 (10.0) | 1747 (35.6) | 817 (16.6) | 439(13.6) | 417 (12.9) | 566(9.6) | 499 (8.5) | 371(9.5) | 296(7.6) | 32(1.3) | 28 (1.1) |
| Primary | 4480 (66.9) | 3983 (59.5) | 2546 (51.8) | 1552 (31.6) | 2316 (71.6) | 2276 (70.3) | 3577 (60.6) | 3200 (54.2) | 1970 (50.5) | 1822 (46.7) | 787(32.1) | 654 (26.7) |
| Secondary and higher | 1419 (21.2) | 1323 (19.8) | 620(12.6) | 565(11.5) | 481(14.8) | 474(14.6) | 1757 (29.8) | 1699 (28.8) | 1564 (40.0) | 1517 (38.9) | 1635 (66.6) | 1534 (62.5) |
| Maternal occupation | | | | | | | | | | | | |
| Not working | 2213 (33.1) | 1910 (28.5) | 2699 (55.2) | 1765 (36.1) | 294(9.1) | 289 (8.9) | 1193 (20.2) | 1082 (18.3) | 1953 (50.0) | 1813 (46.4) | 1381 (56.8) | 1264 (52.1) |
| Professional work | 616(9.2) | 572 (8.6) | 518(10.6) | 415 (8.5) | 434(13.4) | 426 (13.2) | 1235 (21.0) | 1188 (20.1) | 730(18.7) | 710(18.2) | 713(29.4) | 649 (26.7) |
| Nonprofessional work | 3863 (57.7) | 3498 (52.3) | 1670 (34.2) | 738 (15.1) | 2507 (77.5) | 2450 (75.7) | 3469 (58.8) | 3126 (53.0) | 1220 (31.3) | 1113 (28.5) | 335(13.8) | 279 (11.5) |
| Partner education | | | | | | | | | | | | |
| No education | 637(11.4) | 522 (9.3) | 1376 (29.6) | 662 (14.3) | 462(16) | 449 (15.5) | 458(9.2) | 399 (8.0) | 179(6.2) | 149(5.2) | 27(1.3) | 20 (1.0) |
| Primary | 3018(54) | 2735 (48.9) | 2412 (51.9) | 1359 (29.2) | 2051(71) | 2007 (69.5) | 2618 (52.7) | 2342 (47.1) | 1082 (37.5) | 994(34.5) | 483(23.1) | 418 (20.0) |
| Secondary and Higher | 1935 (34.6) | 1803 (32.2) | 856(18.5) | 697 (15.0) | 374(13) | 372 (12.9) | 1896 (38.1) | 1804 (36.3) | 1622 (56.3) | 1555 (53.9) | 1582 (75.6) | 1449 (69.3) |
| Sex of the head of household | | | | | | | | | | | | |
| Male | 5093 (76.1) | 4565 (68.2) | 3474 (70.7) | 1993 (40.6) | 2565 (79.3) | 2516 (77.8) | 4470 (75.7) | 4057 (68.7) | 3111 (79.7) | 2900 (74.3) | 1552 (63.3) | 1403 (57.2) |
| Female | 1600 (23.9) | 1414 (21.1) | 1439 (29.3) | 941(19.2) | 671(20.7) | 650(20.1) | 1431 (24.3) | 1343 (22.8) | 793(20.3) | 735(18.8) | 902(36.7) | 813(33.1) |
| Household size | | | | | | | | | | | | |
| 1–3 | 1329 (19.8) | 1144 (17.1) | 757(15.4) | 422 (8.6) | 663(20.5) | 644(19.9) | 986(16.7) | 901(15.3) | 497(12.7) | 458(11.7) | 490(20.0) | 449(18.3) |
| 4–6 | 3534 (52.8) | 3209 (47.9) | 2515 (51.2) | 1448 (29.4) | 1842 (56.9) | 1809 (55.9) | 2727 (46.2) | 2536 (43.0) | 1729 (44.3) | 1616 (41.4) | 1243 (50.7) | 1130 (46.0) |
| > = 7 | 1829 (27.3) | 1626 (24.3) | 1641 (33.4) | 1064 (21.7) | 731(22.6) | 713(22.0) | 2188 (37.1) | 1962 (33.2) | 1679 (43.0) | 1562 (40.0) | 720(29.3) | 636(25.9) |
| Household wealth index | | | | | | | | | | | | |
| Poorest | 1452 (21.7) | 1278 (19.1) | 1268 (25.8) | 509(10.4) | 645(19.9) | 621(19.2) | 966(16.4) | 852(14.4) | 701(18.0) | 601(15.4) | 561(22.8) | 480(19.6) |

(Continued)

**Table 2.** (Continued)

| Characteristics | Malawi (N = 6693) | | Mozambique (N = 4913) | | Rwanda (N = 3236) | | Uganda (N = 5901) | | Zambia (N = 3905) | | Zimbabwe (N = 2454) | |
|---|---|---|---|---|---|---|---|---|---|---|---|---|
| | N (%) | Yes# n (%) | N (%) | Yes# n (%) | N (%) | Yes# n (%) | N (%) | Yes# n (%) | N (%) | Yes# n (%) | N (%) | Yes# n (%) |
| Poorer | 1374 (20.6) | 1214 (18.1) | 1110 (22.6) | 550(11.2) | 668(20.6) | 652(20.1) | 1097 (18.6) | 965(16.3) | 711(18.2) | 648(16.6) | 544(22.2) | 475(19.3) |
| Middle | 1375 (20.5) | 1209 (18.1) | 992(20.2) | 599(12.2) | 682(21.1) | 670(20.7) | 1189 (20.1) | 1073 (18.2) | 732(18.7) | 679(17.4) | 525(21.4) | 476(19.4) |
| Wealthier | 1259 (18.8) | 1140 (17.0) | 863(17.6) | 662(13.5) | 662(20.5) | 652(20.2) | 1237 (21.0) | 1139 (19.3) | 792(20.3) | 757(19.4) | 441(18.0) | 413(16.8) |
| Wealthiest | 1232 (18.4) | 1138 (17.0) | 679(13.8) | 615(12.5) | 579(17.9) | 571(17.6) | 1412 (23.9) | 1370 (23.2) | 968(24.8) | 950(24.3) | 382(15.6) | 372(15.2) |
| Reads newspaper | | | | | | | | | | | | |
| No | 5601 (83.7) | 4970 (74.2) | 4412 (89.8) | 2503 (50.9) | 2543 (78.7) | 2479 (76.7) | 4778(81) | 4321 (73.2) | 3298 (84.5) | 3055 (78.2) | 1606 (65.5) | 1413 (57.6) |
| Yes | 1091 (16.3) | 1010 (15.1) | 501(10.2) | 431 (8.8) | 689(21.3) | 682 (21.1) | 1123(19) | 1079 (18.3) | 607(15.5) | 581(14.9) | 848(34.5) | 803 (32.7) |
| Listens to the radio | | | | | | | | | | | | |
| No | 3569 (53.3) | 3127 (46.7) | 1705 (34.7) | 977 (19.9) | 600(18.5) | 576 (17.8) | 1623 (27.5) | 1429 (24.2) | 2226(57) | 2058 (52.7) | 1094 (44.6) | 980(39.9) |
| Yes | 3124 (46.7) | 2852 (42.6) | 3208 (65.3) | 1957 (39.8) | 2636 (81.5) | 2590 (80.0) | 4278 (72.5) | 3971 (67.3) | 1679(43) | 1578 (40.4) | 1359 (55.4) | 1235 (50.4) |
| Watches television | | | | | | | | | | | | |
| No | 5656 (84.5) | 5028 (75.1) | 3590 (73.1) | 1883 (38.3) | 2023 (62.6) | 1977 (61.2) | 4288 (72.7) | 3866 (65.5) | 2646 (67.8) | 2419 (61.9) | 1549 (63.1) | 1352 (55.1) |
| Yes | 1037 (15.5) | 951 (14.2) | 1323 (26.9) | 1051 (21.4) | 1207 (37.4) | 1184 (36.7) | 1614 (27.3) | 1533 (26.0) | 1258 (32.2) | 1217 (31.2) | 905(36.9) | 864 (35.2) |
| HIV test uptake [95% CI] | 89.3% [CI: 88.2–90.4] | | 59.7% [CI: 56.2–63.1] | | 97.9% [CI: 97.2–98.3%] | | 91.5% [CI: 90.2–92.7] | | 93.1% [CI: 91.9–94.2] | | 90.3% [CI: 87.8–92.4] | |

N = Total weighted number of eligible women; Yes#n = Weighted number of women who tested for HIV; % = weighted proportions of women who were tested for HIV by study factors in each East African country

higher education, those who resided in Zambia (n = 1517, 38.9%) and Zimbabwe (n = 1534, 62.5%) reported the highest level of usage of HIV test services during pregnancy, whilst lower levels of usage was reported in Burundi (n = 603, 11.2%), Comoros (n = 111, 8.6%) and Ethiopia (n = 296, 6.8%)

## Socioeconomic inequality in prenatal HIV test service uptake in East Africa

Fig 1 illustrates the concentration curve of prenatal HIV test service uptake in East Africa. The concentration curves lay below the 45˚ line (the line of equality), indicating that the prenatal HIV test uptake was more concentrated amongst the rich or higher socioeconomic groups. The overall concentration index for prenatal HIV test service uptake in East Africa was 0.1594 (SE = 0.0042, $P < 0.001$). The positive sign of concentration indices for prenatal HIV test use reveals the existence of pro-rich inequalities. This means women in higher SES utilized more HIV testing services than their counterparts or wealthy women had a higher rate of prenatal HIV testing than those with lower economic status. The magnitude 0.1594 indicates that prenatal HIV test service uptake is 15.94% higher among the rich. Across East African countries, the larger concentration index was measured in Ethiopia with a CI of 0.4033 (SE = 0.0159, $P < 0.001$), while the smallest concentration index was in Rwanda with a CI of 0.0187

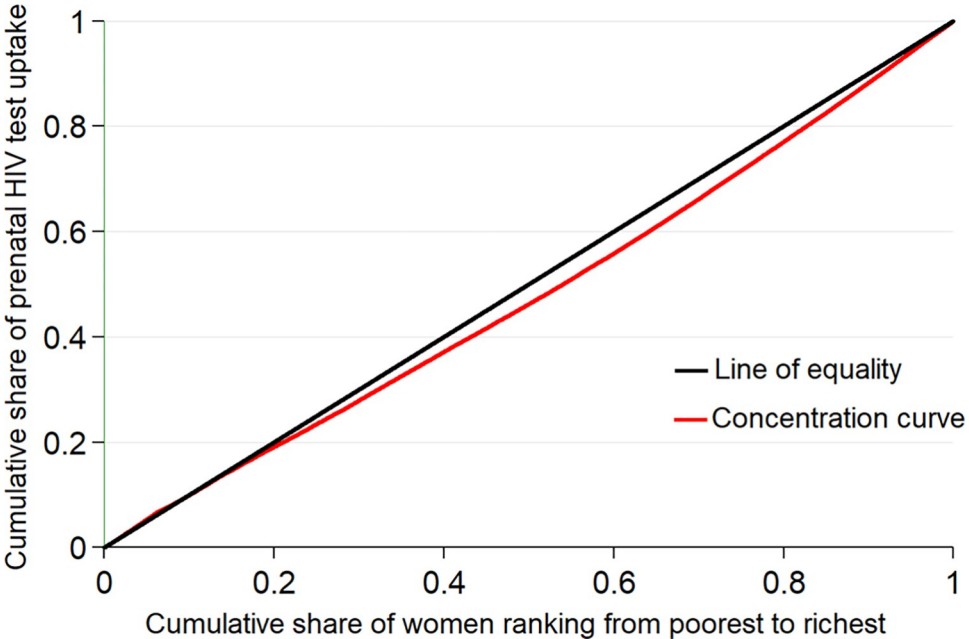

**Fig 1. The concentration curve for prenatal HIV test service uptake for PMTCT in East Africa.**

(SE = 0.0059, $P$ <0.01). This means that prenatal HIV test service uptakes are 40.33% and 1.87% higher among the rich in Ethiopia and Rwanda respectively. The concentration indices and concentration curves across East Africa countries are presented as Supplementary file [S1 File].

### Decomposing socioeconomic inequality in prenatal HIV test service uptake in East Africa

Table 3 and Fig 2 presents the results of the decomposition analysis and the contribution of each explanatory variable to the socioeconomic inequality in prenatal HIV test service utilization. Based on the marginal effect, except for the sex of the household head, all determinants had a significant effect on the likelihood of prenatal HIV testing. For example, women from urban areas had a 4.4% higher chance of being HIV tested compared to those from rural residents. Likewise, more educated women had an increased probability of being HIV tested by 3.6% and 8.3% for primary and secondary/higher level of education respectively as compared with non-educated.

The third column of Table 3 shows the elasticity of prenatal HIV test service uptake for each explanatory variable. The value of elasticity for urban resident women was 0.0388, indicating that a change in women's place of residence from a rural area to an urban area would result in a 3.8% increment in pro-rich socioeconomic inequality of prenatal HIV test service uptake. Likewise, while a change in the country of residence from Burundi to Rwanda (4.0%) increased the pro-rich socioeconomic inequality of prenatal HIV test uptake by 4%, the change in countries of residence from Burundi to Comoros (-5.7%), Ethiopia (-12.2%), Kenya (-1.6%), Malawi (-1.4%), Mozambique (-9.5%), Uganda (-1.6%) and Zimbabwe (-1.8%) reduced the pro-rich socioeconomic inequality in prenatal HIV testing by the percent stated.

The Erreygers concentration index was estimated for each factor related to prenatal HIV test uptake in the fourth column of Table 3. It indicates how the specific variable is distributed

**Table 3. A decomposition analysis of socioeconomic inequalities in prenatal HIV testing for PMTCT in East Africa.**

| Characteristics | Marginal effect | Elasticity | Concentration index (ECI) | Contribution | | |
|---|---|---|---|---|---|---|
| | | | | Value | Percentage | Summed % |
| Countries | | | | | | |
| Burundi | Base | Base | Base | Base | Base | 0.95 |
| Comoros | -0.5056*** | -0.0577 | 0.0018 | -0.0001 | -0.06 | |
| Ethiopia | -0.3245*** | -0.1229 | 0.0208 | -0.0025 | -1.60 | |
| Kenya | -0.0260** | -0.0168 | 0.0895 | -0.0015 | -0.94 | |
| Malawi | -0.0251*** | -0.0147 | -0.0308 | 0.0004 | 0.28 | |
| Mozambique | -0.2219*** | -0.0959 | -0.0594 | 0.0056 | 3.57 | |
| Rwanda | 0.1421*** | 0.0404 | -0.0102 | -0.0004 | -0.25 | |
| Uganda | -0.0311*** | -0.0161 | 0.0270 | -0.0004 | -0.27 | |
| Zambia | -0.0043 | -0.0014 | 0.0159 | -0.00002 | -0.01 | |
| Zimbabwe | -0.0846*** | -0.0182 | -0.0207 | 0.0003 | 0.23 | |
| Residence | | | | | | |
| Urban | 0.0436*** | 0.0388 | 0.4838 | 0.0187 | 11.78 | 11.78 |
| Rural | Base | Base | Base | Base | Base | |
| Maternal age | | | | | | |
| 15–19 years | Base | Base | Base | Base | Base | 0.98 |
| 20–34 years | 0.0149 *** | 0.0267 | 0.0597 | 0.0015 | 1.00 | |
| 35–49 years | 0.0021 | 0.0015 | -0.0254 | -0.00003 | -0.02 | |
| Maternal education | | | | | | |
| No education | Base | Base | Base | Base | Base | 13.69 |
| Primary | 0.0359*** | 0.0751 | -0.1523 | -0.0114 | -7.17 | |
| Secondary and higher | 0.0833*** | 0.0837 | 0.3975 | 0.0332 | 20.86 | |
| Maternal occupation | | | | | | |
| Not working | Base | Base | Base | Base | Base | 4.25 |
| Professional work | 0.0068 | 0.0041 | 0.2251 | 0.0009 | 0.58 | |
| Nonprofessional work | -0.0101* | -0.0195 | -0.3000 | 0.0058 | 3.67 | |
| Partner education | | | | | | |
| No education | Base | Base | Base | Base | Base | 8.24 |
| Primary | 0.0173*** | 0.0343 | -0.1841 | -0.0063 | -3.96 | |
| Secondary and Higher | 0.0400*** | 0.0497 | 0.3911 | 0.0194 | 12.20 | |
| Sex of household head | | | | | | |
| Male | Base | Base | Base | Base | Base | -0.16 |
| Female | 0.0070 | 0.0066 | -0.0395 | -0.0002 | -0.16 | |
| Household size | | | | | | |
| 1–3 | Base | Base | Base | Base | Base | -0.36 |
| 4–6 | 0.0153** | 0.0310 | -0.0217 | -0.0006 | -0.42 | |
| > = 7 | 0.0032 | 0.0042 | 0.0242 | 0.0001 | 0.06 | |
| Household wealth index | | | | | | |
| Poorest | Base | Base | Base | Base | Base | 38.99 |
| Poorer | 0.0203*** | 0.0164 | -0.3548 | -0.0058 | -3.65 | |
| Middle | 0.0395*** | 0.0334 | -0.0230 | -0.0007 | -0.48 | |
| Wealthier | 0.0674*** | 0.0565 | 0.3296 | 0.0186 | 11.69 | |
| Wealthiest | 0.0991*** | 0.0787 | 0.6365 | 0.0501 | 31.43 | |
| Read newspaper | | | | | | |
| No | Base | Base | Base | Base | Base | 2.90 |
| Yes | 0.0286*** | 0.0194 | 0.2378 | 0.0046 | 2.90 | |

(*Continued*)

**Table 3.** (Continued)

| Characteristics | Marginal effect | Elasticity | Concentration index (ECI) | Contribution | | |
|---|---|---|---|---|---|---|
| | | | | Value | Percentage | Summed % |
| Listened to the radio | | | | | | |
| No | Base | Base | Base | Base | Base | 7.11 |
| Yes | 0.0162*** | 0.0375 | 0.3020 | 0.0113 | 7.11 | |
| Watched television | | | | | | |
| No | Base | Base | Base | Base | Base | 7.32 |
| Yes | 0.0224*** | 0.0247 | 0.4725 | 0.0116 | 7.32 | |
| Explained | | | | | | 95.69 |
| Residual | | | | | | 4.31 |

Note: Significance level

***p < 0.001

**p < 0.01

*p < 0.05

across SES. Accordingly, factors such as urban resident (ECI = 0.4838), women aged 20–34 years (ECI = 0.0597), secondary and higher educational level (ECI = 0.3975 for maternal and ECI = 0.3911 for paternal education), higher household wealth index (ECI = 0.3296 for wealthier and ECI = 0.6365 for wealthiest), being exposed to media (ECI = 0.2378 for reading a newsletter, ECI = 0.3020 for listening to the radio, and ECI = 0.4725 for watching TV), had positive Erreygers concentration index, implying that these factors were concentrated amongst women with higher SES or played major contributory roles to inequalities favouring the rich in the use of prenatal HIV test service. The policy implications of these results are that socioeconomic inequality in prenatal HIV testing could be improved by reducing disparity in the distribution of these factors across the economic spectrum, for instance by creating education and media access opportunities to economically disadvantaged groups.

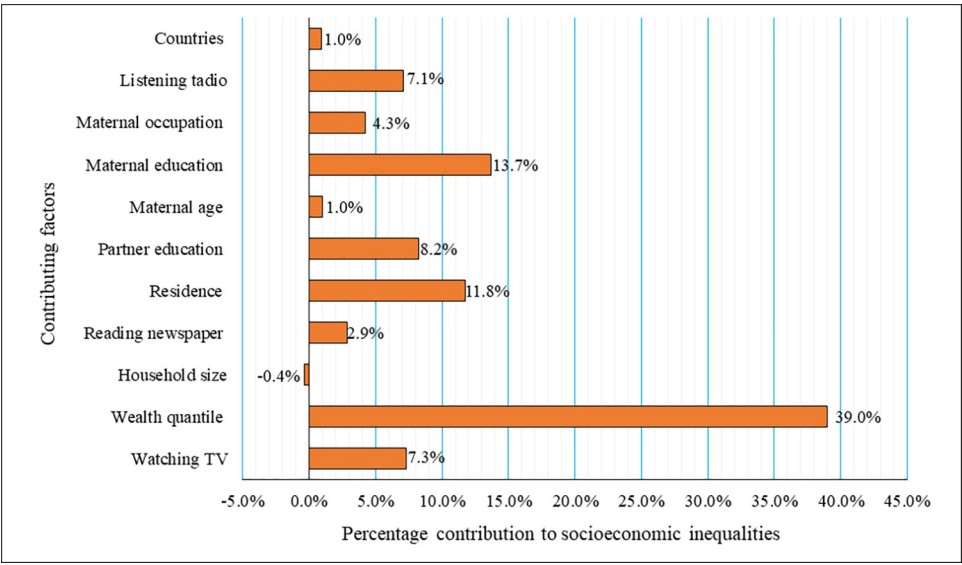

**Fig 2. Contribution of various factors to prenatal HIV test service uptake for PMTCT in East Africa.**

Finally, the fifth and sixth column of Table 3 presents the absolute and percentage contribution of each contributory factor to overall inequality for Prenatal HIV test service uptake. Accordingly, the findings revealed that the major contributors to prenatal HIV test service uptake inequalities in East Africa were household wealth status, maternal education level, and residence. Women from wealthier and wealthiest households explained 43.12% of inequality, followed by women with secondary and higher-level education (20.86%) and urban dwellers (11.78%). Other important contributors to socioeconomic inequality of prenatal HIV test service uptake were partner education (8.24%), listening to the radio (7.11%), and watching television (7.32%). The result furthermore indicates that country of residence, age of woman, and household size played a less important role in determining the prenatal HIV test uptake inequality in East Africa. Overall, included variables in our model explained 95.69% variability of the estimated socioeconomic inequality in prenatal HIV test uptake in East Africa. Unexplained variability or variability explained by factors not included in the model was 4.31% [Table 3].

The contribution of specific factors varied across countries in East Africa. For instance, in Burundi, listening to the radio (68.2%) and maternal occupation (32.9%) contributed most to inequalities in utilization of prenatal HIV test service, whereas in Comoros the greatest contributors were wealth index (32.5%) and partner's education (29.8%). In Ethiopia, wealth index (38.8%) and maternal education (8.3%) were found to be the largest contributors to inequalities in prenatal HIV test service uptake while the same in Mozambique were wealth index (33%) and mother's occupation (14.2%) [S2 File].

## Discussion

Reducing socioeconomic inequality in the utilization of PMTCT services is a major health policy challenge in East African countries. This study showed the existence of socioeconomic inequality in prenatal HIV test service utilization in East Africa with this being 15.94% higher among the rich than the poor (p<0.001). This result is consistent with most studies in developing countries, in which women with higher SES had a high probability of using a range of maternal health services including HIV test services during pregnancy [31, 54–56]. However, the finding contradicts a study in South Africa [39] that found pro-poor inequality of HIV testing during pregnancy among women attending public health facilities. The contradictory finding might be attributable to the fact that underprivileged women are more likely to visit public health facilities than those who are not underprivileged and hence there is a selection bias towards them. Furthermore, the magnitude of pro-rich inequalities in prenatal HIV tests varied across East African countries. The inter-country difference might be the fact that countries involved in this study are East African countries, where political context, economic development, health, and the social policies underway are not the same. It could also be related to variations in the study periods, in countries with the earliest surveys there would be different interventions that have been implemented to improve HIV testing.

Given free-of-charge maternal health services including PMTCT services have been implemented in most East African countries [57–59], the pro-rich inequality of prenatal HIV test service uptake amongst women is a great concern and has policy implications. From this evidence, we can understand that minimizing financial barriers to improve access amongst different socio-economic groups does not automatically result in equitable utilization of the services. Even if the services are provided free of charge or do not involve patient fees, unavailability of services in some areas, and inaccessibility to these services due to costs for transportation [30] could remain a challenge in the region for the poor. Furthermore, studies showed that acceptability and utilization of HIV test services by pregnant women can be negatively

influenced by lack of awareness about MTCT of HIV [18, 60], HIV related stigma and social exclusion [29, 61].

The decomposition analysis of this study identified different significant contributors to socioeconomic inequality in prenatal HIV test services uptake. Place of residence, maternal education, household wealth index, and exposure to media were the variables with the highest contribution to the overall inequality in prenatal HIV test uptake. Women living in urban area were disproportionately rich (positive concentration indices) and they contributed 11.78% to the total observed inequality in prenatal HIV testing in East Africa. Urban women do not face the same barriers to physical access that rural women do; poor roads and greater difficulty of access to health services, low quality and weak integration of PMTCT services with maternal and child health services in rural women, even though services are free [62]. Rural areas are characterized by higher unemployment, lower levels of education, as well as increased poverty. In contrast, in an urban setting, there could be higher employment, education access and successful implementation of the PMTCT programs in terms of services quality, access, and integration [63]. Therefore, ensuring that health facilities are provided quality services and extending the home or community-based HIV testing and counselling programme in rural regions could help to improve access and consequently reduce inequality in HIV testing.

Consistent with other studies [31], a higher education level for both women and their partners contributed 13.69% and 8.24% to this service utilization inequality respectively. Possible explanations of this finding could be that in developing countries, educated partners have higher chance of employment relative to their counterparts, so that employment translates to a better economic situation and can act as an enabling factor for the utilization of healthcare services. Education is also strongly associated with health behaviours and increases individual preventive health care services uptake through improved knowledge, attitudes, and practice. An educated person is more likely to know the risks/consequences of not being tested for HIV during pregnancy [64]. Furthermore, the contribution of secondary and higher education in pro-rich utilization might be explained by people who are more educated being more likely to identify and access affordable healthcare services [18] and make informed decisions about their health [64]. Hence, bridging inequalities in maternal and partner educational levels is an important responsibility to reduce inequalities in the use of prenatal HIV test services.

Many studies have proved that household wealth index is associated with HIV testing [18, 19, 56]. Likewise, in this study, as compared with the poorest groups, the likelihood of HIV testing was 6.7% for women from poorer groups and 9.9% for women from the wealthiest groups. The possible justification may be that pregnant women from the wealthiest households are more likely to receive WHO-recommended antenatal care visits (ANC) (at least four ANC visits) and this would, in turn, increase the chance of receiving HIV testing for PMTCT [65, 66]. Furthermore, previous research has found that there is high self-initiated HIV testing amongst wealthy individuals in countries such as Kenya, Malawi, and Uganda [67]. Based on the decomposition analysis of this study, if the wealth index were equally distributed across women, the socioeconomic-related inequality in prenatal HIV test uptake would have been decreased by 39% or if wealth did not affect the uptake of HIV testing, then uptake of HIV testing amongst the poor would have been increased by 61%.

In this study, women's access to media also significantly contributed to the pro-rich inequality of prenatal HIV test uptake (7.32% for watching television, 7.11% for listening to the radio, and 2.90% for reading newspapers/magazines). This is because the consumption of media is higher amongst wealthier, urban residents and educated women as linked with media infrastructure, electricity access and ownership of private television or radio. Therefore, expanding media coverage or access to radio, television, and newspapers programs amongst women in the lower socioeconomic groups could be an important measure to address the

observed inequality of prenatal HIV test uptake associated with it in East Africa. In this regard, establishing radio stations, distributing radio for mothers in need and broadcasting essential public health information in their local language especially in rural area is recommended. Media can positively influence underprivileged people's health-seeking behaviour by informing, motivating, and reminding them to access healthcare services [68] and therefore, can be used as a strategy to reduce prenatal HIV test service uptake inequality by disseminating well-defined behaviourally focused PMTCT related messages to large audiences, repeatedly and over time [68]. Furthermore, media can be used to improve the general socio-economic and political landscape of any society in the long term.

Our study had several strengths and limitations that need to be acknowledged. To the best of our knowledge, this is the first article that has investigated inequalities in prenatal HIV test uptake from a population-based household survey in East Africa. Evidence provided by this study is important to facilitate the formulation of appropriate strategies to reduce inequalities in prenatal HIV test services uptake in the PMTCT program. The results are comparable across East African countries as all the variables used were similarly described across countries. As a limitation, since this was a cross-sectional study, we were unable to establish any causal relationship between prenatal HIV test service uptake and its determinants. Finally, there is a possibility of recall bias as there is a gap between the time of services use and the timing of data collection.

## Conclusions

We found a pro-rich distribution of prenatal HIV test services utilization in East Africa. However, the performance of the Burundi, Malawi, Rwanda, Uganda, Zambia, and Zimbabwe health systems is promising in narrowing socioeconomic inequalities in prenatal HIV tests for PMTCT. The highest contributors to the inequality of prenatal HIV test service uptake in East Africa included household wealth status, woman's education, access to media and residence. We believe that inequalities observed in this study attributed by these factors are unjust or unfair and are therefore avoidable. The concentration of prenatal HIV test services amongst the high SES groups in East Africa have valuable health policy implications for the elimination of mother-to-child HIV transmission. Implementation of governmental policies and programmes to reduce the gap between the rich and poor (e.g. poverty reduction), higher educational attainment and HIV testing targeting socio-economically disadvantaged women (e.g. Home/community-based HIV testing and counselling) are highly recommended.

## Supporting information

**S1 File. Erreygers' normalized concentration indexes and curves for prenatal HIV test uptake across East African countries.**
(DOCX)

**S2 File. A decomposition analysis of socioeconomic inequalities in prenatal HIV testing for PMTCT among mother aged 15–49 years in Burundi and Comoros.**
(DOCX)

## Acknowledgments

The authors would like to thank the DHS program for collecting and making available the data and the women who participated in these surveys. We also extend our deepest gratitude to Western Sydney University for allowing the conduct of this study.

## Author Contributions

**Conceptualization:** Feleke Hailemichael Astawesegn.

**Data curation:** Feleke Hailemichael Astawesegn.

**Formal analysis:** Feleke Hailemichael Astawesegn, Haider Mannan.

**Investigation:** Feleke Hailemichael Astawesegn, Elizabeth Conroy, Haider Mannan, Virginia Stulz.

**Methodology:** Feleke Hailemichael Astawesegn, Elizabeth Conroy, Haider Mannan, Virginia Stulz.

**Project administration:** Feleke Hailemichael Astawesegn, Elizabeth Conroy.

**Supervision:** Elizabeth Conroy, Haider Mannan, Virginia Stulz.

**Visualization:** Feleke Hailemichael Astawesegn, Elizabeth Conroy, Haider Mannan, Virginia Stulz.

**Writing – original draft:** Feleke Hailemichael Astawesegn, Elizabeth Conroy, Haider Mannan, Virginia Stulz.

**Writing – review & editing:** Feleke Hailemichael Astawesegn, Elizabeth Conroy, Haider Mannan, Virginia Stulz.

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
