## [Decision Letter · Decision Letter 0]

11 Apr 2022

PONE-D-21-35264

Measuring socioeconomic inequalities of prenatal HIV test uptake for prevention of mother to child transmission of HIV in East Africa: A decomposition analysis

PLOS ONE

Dear Dr. Astawesegn,

Thank you for submitting your manuscript to PLOS ONE. After careful consideration, we feel that it has merit but does not fully meet PLOS ONE’s publication criteria as it currently stands. Therefore, we invite you to submit a revised version of the manuscript that addresses the points raised during the review process.

I would like to sincerely apologise for the delay you have incurred with your submission. It has been exceptionally difficult to secure reviewers to evaluate your study. We have now received two completed reviews; the comments are available below. The reviewers have raised significant scientific concerns about the study that need to be addressed in a revision.

Please revise the manuscript to address all the reviewer's comments in a point-by-point response in order to ensure it is meeting the journal's publication criteria. Please note that the revised manuscript will need to undergo further review, we thus cannot at this point anticipate the outcome of the evaluation process.

We look forward to receiving your revised manuscript.

Kind regards,

Miquel Vall-llosera Camps

Senior Editor

PLOS ONE

3. We noticed you have some minor occurrence of overlapping text with the following previous publication, which needs to be addressed:

- https://pubmed.ncbi.nlm.nih.gov/34065689/

The text that needs to be addressed involves the Materials and Methods section.

In your revision ensure you cite all your sources (including your own works), and quote or rephrase any duplicated text outside the methods section. Further consideration is dependent on these concerns being addressed.

Reviewers' comments:

Reviewer's Responses to Questions

**Comments to the Author**

1. Is the manuscript technically sound, and do the data support the conclusions?

Reviewer #1: Yes

Reviewer #2: Partly

2. Has the statistical analysis been performed appropriately and rigorously? 

Reviewer #1: Yes

Reviewer #2: No

3. Have the authors made all data underlying the findings in their manuscript fully available?

Reviewer #1: Yes

Reviewer #2: Yes

4. Is the manuscript presented in an intelligible fashion and written in standard English?

Reviewer #1: Yes

Reviewer #2: Yes

5. Review Comments to the Author

Reviewer #1: Summary:

The authors investigated wealth-related socio-economic inequality in prenatal HIV testing, in 10 East African countries using DHS data. They report a pro-rich inequality index indicating better prenatal HIV testing amongst pregnant women of higher socioeconomic status. They also found that place of residence, maternal education, household wealth index, and exposure to media contributed significantly to the overall inequality of prenatal HIV test uptake. Urban dwellers and higher educated women for example had better uptake of HIV testing. The gap between the rich and poor with respect to access and utilization of antenatal healthcare services needs to be dealt with urgently in this region.

The study is very well written and methodologically sound. I have essential but not major revision recommendations outlined below.

Minor Essential Revisions:

OVERALL:

- Please explain abbreviations at the first time you use them. E.g. MTCT in the abstract, PMTCT & SES in the introduction. are not explained the first time they are mentioned. Terms explained already in the introduction should then use abbreviations thereafter – I realize you still used “prevention of mother-to-child transmission of HIV” under Materials & methods.

- The phrase “PMTCT of HIV” can be avoided if you define ‘PMTCT’ to include HIV, but you have not defined PMTCT.

- Run the manuscript through a plagiarism score check

ABSTRACT: Useful to indicate the period of data collection in years.

INTRODUCTION: I would re-order the paragraphs on the Introduction:

- move paragraphs 3,4,5 to be the first, second & third paragraphs, respectively as these nicely introduce the main health issue and then zoom into the specific topic of the paper. Here I recommend listing the East African countries which were included in the 81% estimate (I recommend this because some countries like Mozambique, Zambia & Zimbabwe are sometimes grouped under Southern Africa in global statistics reports).

- The fourth paragraph would start with the sentence: “Measuring socioeconomic inequalities in health service use can guide the focus of researchers and assist policymakers to prioritize specific health care needs (38)” . This way you can reduce current paragraphs 1 & 2 into a single paragraph (new paragraph 4) to be less broad but focus on defining socioeconomic inequalities and how they are measured. This paragraph would touch on one or two examples of methods for measuring these inequalities – thus an opportunity to introduce important terms such as “decomposition analysis’ which you mention in your objectives paragraph, and ‘concentration indices’ which you mention in the methods, but both not explained there. Some sentences can be easily moved from the Methods section, e.g. “CI is a popular tool to measure health and health care inequality in the field of health policy and health economics research (56).” AND “The concentration index (CI) was used to measure the extent/magnitude of socioeconomic inequalities in prenatal HIV test service utilization (54, 55).” The current first two paragraphs have some useful content but some of it is not necessary and makes the intro wander away slightly.

- You may end with your objectives paragraph.

RESULTS: It might be useful to the reader in understanding why some percentages do not add up to 100%, if you indicate with a character like * in Table 1 those variable/country cells where there was missing information. E.g., Burundi ‘Urban = 8.7% and rural = 81.0%.

DISCUSSION: Could some of the inter-country differences somehow influence the differences in results between countries? I would imagine the surveys conducted in 2011-2012 (Comoros & Mozambique) would be not comparable to those conducted in 2016 and beyond given the change in PMTCT guidelines over time. Authors may review changes in PMTCT guidelines around prenatal HIV testing across the survey years and discuss whether this could have impacted on the survey data. The authors may decide whether this is a simple discussion point or a limitation.

Discretionary Revisions:

At the author’s choice – would using South-East and East Africa be more accurate given the geographical location of Mozambique, Zambia and Zimbabwe is accurately South-East Africa and the latter two are commonly grouped under Southern Africa as well.

Reviewer #2: 1. I would suggest focusing the intro on the topic and not on explaining what are inequalities and inequities. That could be a paragraph with references.

2. The intro could explain why the topic is relevant and we already know about it. As the authors stated, there is already evidence of differences by socioeconomic status in HIV testing, so what is the value-added of characterizing those using the CI? How this metric could contribute to informing better policies? That is, for example, knowing that income level is a relevant contributor to testing inequalities could modify policies?

3. In terms of the data, how using a period of 7 years of surveys could affect the analysis? That is, during this period (2011-2018), there were interventions to improve testing in those countries, in particular in the countries with the earliest surveys?

4. How comparable is the wealth index across countries? That is, the metric is relative to households in the country, not between countries. Also, my understanding is that the WI uses a set of variables that could differ county by country as some assets are less or more relevant depending on the context.

5. The WI was used as a continuous measure or the quintiles?

6. There is a ? mark on formula 3 and the text that follows

7. It seems to me that the authors interpretation of the CI is not adequate. The CI is a measure of the degree of inequality, that is, how concentrated is the health indicator among individuals with higher (lower) socioeconomic level. The value of the CI is not a percentage difference between two categories (rich/poor) as it is estimated on the complete spectrum of SE level.

8. The interpretation on the decomposition also seems not adequate. The coefficients are the relation between the CI for testing and the CI for the explanatory variable, so indicates how the inequality on the explanatory variable is related to inequality in testing.

9. Interpretations as a change of residence to decrease inequalities seem not really informative. It would be important to discuss this from social determinants of health perspective. Suggesting that distributing radios —for example— could improve equality on testing may be missing the structural factors that may be related to listening to the radio in a particular context.

10. The discussion could be improved by discussing the findings using the framework of the social determinants of health, rather than trying to discern how listening to the radio or watching tv are related to testing.

6. PLOS authors have the option to publish the peer review history of their article (what does this mean?). If published, this will include your full peer review and any attached files.

Reviewer #1: No

Reviewer #2: **Yes: **Juan Pablo Gutierrez

---

## [Author Response · Author response to Decision Letter 0]

21 Jun 2022

I have attached one by one response to the editor and reviewers as a separate file.

---

## [Decision Letter · Decision Letter 1]

10 Aug 2022

Measuring socioeconomic inequalities in prenatal HIV test service uptake for prevention of mother to child transmission of HIV in East Africa: A decomposition analysis

PONE-D-21-35264R1

Dear Dr. Astawesegn,

We’re pleased to inform you that your manuscript has been judged scientifically suitable for publication and will be formally accepted for publication once it meets all outstanding technical requirements.

Kind regards,

Joseph Donlan

Staff Editor

PLOS ONE

Additional Editor Comments (optional):

Reviewers' comments:

Reviewer's Responses to Questions

**Comments to the Author**

1. If the authors have adequately addressed your comments raised in a previous round of review and you feel that this manuscript is now acceptable for publication, you may indicate that here to bypass the “Comments to the Author” section, enter your conflict of interest statement in the “Confidential to Editor” section, and submit your "Accept" recommendation.

Reviewer #1: All comments have been addressed

2. Is the manuscript technically sound, and do the data support the conclusions?

Reviewer #1: Yes

3. Has the statistical analysis been performed appropriately and rigorously? 

Reviewer #1: Yes

4. Have the authors made all data underlying the findings in their manuscript fully available?

Reviewer #1: Yes

5. Is the manuscript presented in an intelligible fashion and written in standard English?

Reviewer #1: Yes

6. Review Comments to the Author

Reviewer #1: The authors have addressed all my comments.

One minor error needs to be noted.

ABSTRACT: The ‘MTCT’ abbreviation can be introduced in the first sentence of the ABSTRACT where authors mention ‘mother-to-child transmission of HIV’, and then simply used in the ‘Conclusion’

The authors may consider indicating in the ABSTRACT the years for which the reported data apply.

7. PLOS authors have the option to publish the peer review history of their article (what does this mean?). If published, this will include your full peer review and any attached files.

Reviewer #1: No

---

## [Editor Report · Acceptance letter]

12 Aug 2022

PONE-D-21-35264R1 

Measuring socioeconomic inequalities in prenatal HIV test Service uptake for prevention of mother to child transmission of HIV in East Africa: A decomposition analysis 

Dear Dr. Astawesegn:

I'm pleased to inform you that your manuscript has been deemed suitable for publication in PLOS ONE. Congratulations! Your manuscript is now with our production department. 

Kind regards, 

on behalf of

Dr Joseph Donlan 

Staff Editor

PLOS ONE